# Collaborative Study for Iodine Monitoring in Mandatory Direct-Iodized Sauce in Thailand

**DOI:** 10.3390/foods12183513

**Published:** 2023-09-21

**Authors:** Juntima Photi, Kunchit Judprasong, Sueppong Gowachirapant, Premmin Srisakda, Jutharat Supanuwat, Christophe Zeder

**Affiliations:** 1Institute of Nutrition, Mahidol University, Salaya, Phuthamonthon, Nakhon Pathom 73170, Thailand; juntima.pho@mahidol.ac.th (J.P.); sueppong.gow@mahidol.ac.th (S.G.); premmin.sri@mahidol.ac.th (P.S.); 2Bureau of Nutrition, Department of Health, Ministry of Public Health, Nonthaburi 11000, Thailand; s_jutharat@hotmail.com; 3Laboratory of Human Nutrition, Department of Health Sciences and Technology, Institute of Food, Nutrition, and Health, ETH Zurich, Schmelzbergstrasse 7, 8092 Zurich, Switzerland; christophe.zeder@hest.ethz.ch

**Keywords:** iodine deficiency, iodine analysis, inductively coupled plasma–mass spectrometry, fish sauce, soy sauce, seasoning sauces

## Abstract

Direct iodization in fish sauce, soy sauce, and seasoning sauces plays a crucial role in optimizing the iodine intake of Thailand’s people. However, determining the iodine content to ensure that these sauces meet the standard of Thailand’s Food and Drug Administration (FDA) is challenging. In this study, all local laboratories equipped with inductively coupled plasma–mass spectrometry (ICP-MS) and with experience in iodine analysis by any analytical method were invited to participate in a hands-on training workshop and two rounds of interlaboratory comparison. The aim was to improve laboratory performance and assess the potential for iodine monitoring for mandatory direct-iodized sauces. All target laboratories participated in this study. The hands-on training workshop harmonized the analytical method and increased the capacity of participating laboratories. Most laboratories (7/8) achieved satisfactory performance for six test samples based on interlaboratory comparison. Samples were extracted by tetramethylammonium hydroxide (TMAH), with the presence of 6% 2-propanol, 0.01% triton X-100, internal standard, and iodine determination in direct-iodized sauces by ICP-MS. The reproducibility standard deviation (*S_L_*), after the removal of outlier results for iodine content, was 7–22% iodine at a level of 0.03–4.81 mg/L. Moreover, the Thai FDA’s judgment range for official control activities should expand the range of 2–3 mg per 1 L (ppm) by at least 22%.

## 1. Introduction

Iodine is an essential micronutrient for producing thyroid hormones that are crucial for cell metabolism, reproduction, and growth. Iodine deficiency is the main public health problem associated with thyroid dysfunction, leading to goiters, cretinism, the development of abnormalities, intellectual disability, and increased perinatal mortality [1,2]. Since 1994, the World Health Organization (WHO) and the United Nations Children’s Fund (UNICEF) have been promoting universal salt iodization (USI) to eliminate iodine deficiency worldwide [3]. After two decades of USI implementation, global perspectives on iodine status in 2020 found that the populations of most countries (118 out of 152, including Thailand) have achieved adequate iodine intake. However, other countries have either deficient or excessive intakes (21 and 13 from 152 countries, respectively) [4].

In Thailand, the first notification for mandatory salt iodization at a level not less than 30 mg iodine per 1 kg (ppm) of table salt was announced in 1994 [5]. Thailand then adopted USI in 1999 and enacted a notification on iodized salt in 2010 that extended the scope to include edible salt used in food or used as a mixture or an ingredient in food in order to increase the efficiency of the iodine deficiency reduction program [6,7]. However, Chavasit et al. found that after 12 months of fermentation, the loss of iodine varied between 13 and 55% in fish sauce produced using standard iodized salt (approximately 30 ppm iodine) as an ingredient, and residual iodine in the final product was up to 80–143 μg per serving (15 mL) [8,9]. Moreover, the Thai people usually consume other direct-iodized sauce, such as fish sauce, soy sauce, and seasoning sauces, in greater quantities than for just table salt alone [10,11]. Consequently, Thailand modified the USI concept according to the country’s direct-iodized sauce preference and unique cultural context. Hence, the Thai Food and Drug Administration (FDA) allowed for the direct addition of iodine in the range of 2–3 mg per 1 L (ppm), or approximately 30–45 μg per serving, to fish sauce, soy sauce, and seasoning sauces, as well as using iodized salt as an ingredient. The notification on edible salt was also revised to mandate that edible salt should have an iodine content between 20 and 40 ppm [9,12,13,14,15].

Salt iodization requires an efficient machine to homogeneously mix iodine into salt. Moreover, the direct addition of iodine solution into a sauce does not change the color or taste of the product and does not require any specific technology [16,17]. Consequently, it can save tremendously on fortification costs. Either potassium iodide (KI) or potassium iodate (KIO_3_) can be used as the iodine fortificant. However, KIO_3_ is more stable and can be used alone for any type of salt quality, while KI is used for salt of very good quality and using other substances to stabilize KI [18]. Moreover, iodine in direct-iodized sauce with KIO_3_ is quite stable. The loss of iodine in direct-iodized fish sauce using potassium iodide (KI) or potassium iodide (KIO_3_) after 3-month storage is less than 18.5% and 0%, respectively [17].

The estimated iodine intake of Thai non-pregnant adults in 2021 from household salt and 12 processed foods contributed the most to daily iodine intake when using iodized salt in all suitable products, reaching up to 147% of the recommended nutrient intake (RNI) for iodine. The daily iodine intake when iodized salt was used in nine products with and without direct iodization (2–3 ppm) as per fish sauce, soy sauce, and seasoning sauces was approximately 116% and 89% of the RNI [19]. Consequently, directly adding iodine to fish sauce, soy sauce, and seasoning sauces has a crucial role to play in achieving optimal iodine intake in the Thai population.

Iodine determination for those directly iodized sauces, however, is challenging, since the fortification concentration is quite low, and iodine tends to be easily reduced and oxidized. Several analytical methods have been developed for analyzing and measuring iodine in any food sample matrix. Among them, inductively coupled plasma–mass spectrometry (ICP-MS) detection can provide excellent selectivity and sensitivity [20,21,22]. The iodine content from the ion-selective electrode (ISE) tends to be lower than the value from ICP-MS because ISE is sensitive to iodide, while the ICP-MS technique takes all forms of iodine into account [23]. Moreover, the iodine content of foods containing iodine at more than 0.25 ppm, as determined by spectrophotometry, is higher than the value from ICP-MS, which varies from 25% to 122% [24]. The ICP-MS technique for iodine determination in food has now been published as an international standard and noted in the Elemental Analysis Manual (EAM) for food and related products of the U.S. FDA [25,26]. In Thailand, two laboratories (one government and one private) are known to provide iodine-determination services in foods by using ICP-MS, but variations in their test results are problematic when using the Thai FDA standard range for direct-iodized sauces. As a case in point, Chavasit and Photi in 2020 (unreported data) found that the iodine levels determined by ICP-MS in the same fish sauce sample as reported by three different laboratories (the two mentioned earlier and one internationally recognized laboratory from abroad) were 3.29, 1.88, and 2.70 ppm. The results differed by 3–28% from an average of 2.62 and were not in the same direction. One was stated to meet the standard (2–3 ppm), while the others were above or below the standard. Consequently, discrepancies in analytical results can mislead regulators and policymakers and can potentially cause unfair legal actions against food industries.

Improving the capabilities of Thailand’s laboratories is required in order to sustain the mandatory iodized program for direct-iodized sauces and optimize the iodine intake of the population. In addition, limitations in iodine monitoring under the present standard of the Thai FDA must be assessed. Potential improvements in the monitoring program should also be specified and integrated into the legal framework.

## 2. Materials and Methods

### 2.1. Hands-on Training Workshop

A hands-on training workshop in iodine analysis in food using ICP-MS was organized in Thailand by the Institute of Nutrition, Mahidol University (INMU), in cooperation with an expert from the Institute of Food, Nutrition and Health, the Swiss Federal Institute of Technology Zurich (ETH Zurich). A list of laboratories in Thailand was compiled from an information database for food entrepreneurs about agencies registered with the Thai FDA or FDA that had acceptable analysis results [27]. In addition, the available websites of those laboratories were browsed with the keywords “Iodine” and “ICP-MS” to identify those equipped with ICP-MS and having experience in iodine analysis using it. The National Institute of Metrology of Thailand (NIMT) laboratory was also classified as a target laboratory. All potential and target laboratories were invited to participate in the hands-on training workshop.

During the hands-on training workshop conducted from 21 to 23 March 2022, a senior chemist from the Human Nutrition Laboratory, ETH Zurich, who had expertise in iodine determination gave a lecture on iodine analysis using the ICP-MS method (extraction, digestion, and determination) and the critical control points for each step. On the following day, participants practiced iodine determination under the supervision of senior chemists from ETH and INMU, using samples of non-iodized fish sauce (Rayong Fish Sauce Industry Co., Ltd., Rayong, Thailand), non-iodized fish sauce containing an iodine standard (TraceCERT^®^, 1000 mg/L iodide in water, Merck Ltd., Darmstadt, Germany), and SRM 1869 Infant/Adult Nutritional Formula II (National Institute of Standards and Technology: NIST). The steps of sample extraction (acid or alkaline digestion), standard preparation, determination of iodine content using ICP-MS, and data analysis were performed in triplicate. In addition, alkaline digestion and ICP-MS condition followed the method published by Todorov and Gray in 2016 [22].

### 2.2. Collaborative Interlaboratory Comparison

Two rounds of interlaboratory comparison for iodine analysis in iodized and non-iodized sauces were organized by INMU in March 2020 (round 1) and February 2023 (round 2). All participating laboratories from the hands-on training workshop and an ETH laboratory, which was identified as a reference laboratory, were invited to participate. An invitation letter provided relevant information, instructions, and reporting tables. All participants were anonymized with confidential code numbers. In addition, the participating laboratories were requested to submit a report of triplicate test results in the unit of mg/L (ppm) with at least two decimal digits within two months of receiving the test materials.

#### 2.2.1. Test Materials Preparation

Iodized and non-iodized fish sauce, soy sauce, and seasoning sauce samples were commercial products purchased from supermarkets and factories within a week before organizing each round of interlaboratory comparison. The iodine solution (480 ppm) was prepared by dissolving 4 g of potassium iodate (KIO_3,_ Calibre Chemicals Pvt. Ltd., Gujarat, India) in 100 mL of DI water and then diluted 50 times with DI water. During sample preparation, the prepared iodine solution was added to some commercial products at the ratio of 3.5 mL of iodine solution per liter of sauce to cover iodine content approximately two times the standard range in the FDA notification (Table 1). Two liter of the sample from a single bulk package was mixed using a magnetic stirrer (IKA^®^, C MAG HS7, IKA Works Inc., Wilmington, DE, USA) for 15 min and repacked into individual 50 mL bottles (50 mL × 40 bottles). Ten test samples were selected by stratified random sampling for homogeneity testing, and duplicate sample portions from each bottle were tested using ICP-MS prior to dispatch. Stability testing was conducted at the end of the study (2 months) by selecting 4 test materials using simple random sampling and analyzing iodine content by ICP-MS. Immediately after batch preparation, randomly selected containers were stored at room temperature (25 ± 5 °C) followed by determination of iodine content using ICP-MS. Samples were distributed to participating laboratories under sunlight-protected packages. The aim was for the laboratories to receive their samples within 2 weeks after preparation.

#### 2.2.2. Statistical Analysis

Statistical analysis for homogeneity testing was conducted based on ISO 13528:2022 [28]. After passing the Cochran’s test (test for outliers for duplicate results), the between-sample standard deviation (*s_s_*) of each test material was calculated. Values should be less than 0.3 times the standard deviation for proficiency testing ($s_{s}\leq 0.3\sigma_{pt}$), which is considered to be adequately homogeneous. The standard deviation for the PT of each sample was derived from Horwitz’s equation [28]. If this criterion was not met, the expand criterion ($c=F_{1}\sigma_{allow}^{2}+F_{2}s_{w}^{2}$) was applied to allow for an actual sampling error and repeatability in the homogeneity study [28]. If *s_s_* < $\sqrt{c}$, the proficiency test items were considered to be sufficiently homogeneous.

The end-of-study stability results were evaluated using the criteria of ISO 13528:2022 [28], which compared the mean value of the iodine at the end of study (${\bar{y}}_{1}$) to the mean value of homogeneity testing (${\bar{y}}_{1}$). If $\left|{\bar{y}}_{1}-{\bar{y}}_{2}\right|\leq 0.3\sigma_{pt}$, the proficiency test items were considered to be adequately stable.

The laboratories’ results were analyzed and interpreted by the organizer of this collaborative study who was not involved in performing iodine determination for any test sample in order to prevent any conflict of interest. For each study round, the test results from all participating laboratories were analyzed to establish assigned values according to ISO 13528:2022 [28]. Firstly, a scatter plot and kernel density plot were used to determine the distribution of all submitted results. Grubb’s test and Dixon’s test were used to identify outliers. Grubbs’ test (G test) is used to identify an outlier in a data set if a minimum value or a maximum value is an outlier ((suspected result − mean)/standard deviation) compared to the critical value at 95% confidence. If the calculated G value is larger than the critical G value at 95% confidence, it is defined as an outlier. Dixon’s test (Q test) is also used to identify and reject outliers. To apply a Q test for suspected data, the gap in the closeness of data was divided by the range (max–min) of data (Q = gap/range) after sorting as per increasing data. If the calculated Q value is more than the critical Q value at 95% confidence, it is defined as an outlier. Meanwhile, Cochran’s test at 95% confidence was used for intralaboratory evaluation (within-laboratory). Cochran’s test is used to identify a single estimate of variance compared to group variances. If the calculated Cochran value is larger than the critical Cochran value at 95% confidence, it is defined as an outlier.

For between-laboratory evaluation, test results for each laboratory (*x_i_*) were evaluated for laboratory performance, using a *z*-score (Equation (1)), against two approaches, including reference values from ETH laboratory (*x*_pt_) and standard deviation based on Horwitz’s RSD (*σ*_pt_). Another approach using the consensus mean (*x*_pt_) and standard deviation (*σ*_pt_) of all participants after removing outliers was also used for performance evaluation. The *z*-score against assigned values was evaluated for between-laboratory variations as follows (Equation (1)):(1)$$z-score=\frac{\left(x_{i}-x_{\mathrm{pt}}\right)}{\sigma_{\mathrm{pt}}}$$
where *x*_i_ is the average value reported by each participating laboratory; *x*_pt_ is the assigned mean value of reference laboratory or consensus mean after removing the outlier; and σ_pt_ is the standard deviation PT, as estimated by Horwitz’s equation [28], or the consensus SD after outlier removal. The uncertainty of assigned value ($u_{x_{pt}}$) was calculated by the standard deviation of all submitted results after removing outliers (SD) and divided by the square root of the number of participating laboratories (*p*) ($u_{x_{pt}}=SD/\sqrt{p}$). If the standard uncertainties of the assigned values in this study were smaller than $0.3\sigma_{pt}$, then the uncertainty of the assigned value was negligible and not included in the interpretation of the evaluation results in this study.

Laboratories with the |*z*| for ≤2 for between-laboratory variations were considered to have a satisfactory performance. Those with 2 < |*z*| < 3 or |*z*| ≥ 3 presented questionable or unsatisfactory results, respectively. The summary of each round of collaborative study was reported to all participating laboratories in a technical meeting for each round.

For repeatability and reproducibility of the standard deviation for the group of laboratories, they were calculated using one-way ANOVA based on the EURACHEM guideline [29]. The repeatability standard deviation (*S_r_*) was calculated by taking the square root of the within group (*MS_w_*). The reproducibility standard deviation (*S_R_*, s_between_) was calculated from the different of mean square between group (*MS_b_*) minus within group (MS_w_) divided by the number of replicate (*n*) as follows (Equation (2)):(2)$$S_{R}=\sqrt{\frac{{MS}_{b}-{MS}_{w}}{n}}$$

## 3. Results

### 3.1. Homogeneity and Stability Testing

The results for the homogeneity and stability of the test materials are summarized in Table 2. Most test materials demonstrated sufficient homogeneity; the MU-ETH I04 test material was only adequately homogeneous. In addition, all test materials were adequately stable after storing for 2 months (end of study). These indicated that all test materials were suitable for interlaboratory comparison.

### 3.2. Results of the Hands-on Training Workshop

Twenty-one staff members from all target laboratories in Thailand (Table 3) participated in the hands-on training workshop. The test results for iodine determination using ICP-MS during the workshop revealed that the result of the NIST SRM 1869 (infant formula) from alkaline extraction passed the accuracy test (*t_calculate_* = 0.824; and *t_critical_* = 4.303). However, the test result for acid microwave digestion did not pass the criteria (*t_calculate_* = −19.842; and *t_critical_* = 4.303). Consequently, higher precision was obtained from alkaline extraction (4.1% RSD) compared to 28% RSD by microwave digestion. The recoveries of the iodine-spiked samples were 90.8 ± 7.1% (*n* = 3, 7.9% RSD) and 27.0 ± 1.66% (*n* = 3, 6.2% RSD) in sample dissolution by alkaline extraction and acid microwave digestion, respectively. The recovery of the alkaline extraction stayed within an acceptable range (80–110% at the level of 800 µg/kg) [30]. The method limit of detection (3 SD of 10 analysis of lowest concentration) and method limit of quantitation (10 SD of 10 analysis of lowest concentration) were 0.06 and 0.20 mg/L, respectively.

### 3.3. Interlaboratory Comparison

#### 3.3.1. Participating Laboratories

Nine laboratories participated in the collaborative study for interlaboratory comparison, including all seven target laboratories, the laboratory that represented the collaborating group of government, and the private laboratories during the hands-on training workshop, as well as the ETH’s laboratory. Eight laboratories participated in each round of the collaborative study, all of which used the alkaline extraction method for extracting iodine content from the test samples. The exact amounts of samples (0.5–2.0 g) and 4–5% tetramethylammonium hydroxide (TMAH) were mixed and extracted in an oven at about 90 °C for 3 h. For the ICP-MS condition, 10 ppb tellurium (Te) or 2 ppb rhodium (Rh) in 1% TMAH, 0.01% triton X-100, and 6% 2-propanol were used as internal standards. The details of each laboratory’s method of analysis and instruments are shown in Table 4 and Table 5, respectively.

#### 3.3.2. Assigned Iodine Content

The results of the collaborative study’s first round showed that all participating laboratories passed all the outlier tests. The mean iodine contents in sample codes MU-ETH I04, I05, and I06 were 4.81, 0.27, and 2.10 mg/L, respectively. In the second round, however, the test results for each sample indicated that one of the eight laboratories did not pass the kernel density plot and a scatter plot. Moreover, one of the eight laboratories did not pass the Cochran’s test at 95% confidence in the case of sample codes MU-ETH I07 and I09. Kernel density plots for all test materials with the suitable bandwidth at 0.75 σ_pt_ [28] for all data and data removing outliers are shown in Figure 1. The results clearly indicate the unimodality (normal distribution) of each assigned value. Consequently, after removing the outliers, the mean iodine content in sample codes MU-ETH I07, I08, and I09 were 2.10, 0.088, and 0.032 mg/L, respectively (Table 6). The standard uncertainties of the assigned values for each test material were smaller than 0.3σ_pt_. Consequently, the *z*-score was used for a performance evaluation.

#### 3.3.3. Performance of Participating Laboratories

Figure 2 shows the combined laboratory performance from two rounds of the collaborative study when the *z*-score is based on the reference lab as ETH’s mean and standard deviation from the Horwitz equation. It indicates that the test results of all samples from lab codes 01–07 showed satisfactory results (|*z*| ≤ 2.0). Meanwhile, the test results of lab code 08 in test sample MU-ETH I07 and lab code 09 in two test samples (MU-ETH I08 and MU-ETH I09) were unsatisfactory (|*z*| ≥ 3.0). Figure 3 shows the laboratory performance when the *z*-score is based on the consensus-assigned mean and standard deviation, and the interpretation of the plot is similar to that of Figure 2. The test results of the two collaborative study rounds showed the same pattern. The standard deviation from all laboratories after outlier removal indicated that the repeatability standard deviation (*S_r_*) varied from 2 to 17%, while the reproducibility standard deviation (*S_L_* or *S_R_*) ranged from 7 to 22% (Table 7).

## 4. Discussion

Hands-on training workshop results showed that alkaline extraction by TMAH provides higher accuracy and precision for iodine determination in food compared to acid digestion. These results support the finding of Lehner et al. [31] that indicated that iodine recoveries from the alkaline method were significantly higher than those from acid digestion. Consequently, all participating laboratories in this collaborative study used the alkaline extraction method for interlaboratory comparison. The hands-on training workshop could increase the capacity of service laboratories for iodine analysis in food, since the *z*-score and the repeatability standard deviation (*S_r_*) for all participating laboratories indicated that most laboratories achieved a satisfactory performance. Alkaline extraction by TMAH and ICP-MS detection with 6% 2-propanol and 0.01% triton X-100 in an internal standard can be used for iodine determination in direct-iodized sauces. The adjusted standard deviation of reproducibility (*S_L_*) after removing the outliers from all laboratories with that iodine determination method stayed within the acceptable range of AOAC International (*S_L_* = 11%, 16%, and 22% for concentrations of 10 ppm, 1 ppm, and 100 ppb, respectively) [30]. Alkaline extraction and ICP-MS detection have been used to determine the iodine content of soy sauces sampled in China [26,32]. In addition, in Taiwan, the ICP-MS method with sample preparation by diluting the sample 100-fold into an aqueous solution containing Triton X-100 and 0.5% ammonia solution has been used as the reference method for validating the modified microplate method to measure iodine in soy sauces [33,34].

The unacceptable *z*-score of some laboratories and high *S_L_* (>35%) were found in sample codes MU-ETH I08 and MU-ETH I09 due to a small amount of iodine (<100 ppb) in the test samples. That amount is below the method limit of quantification (LOQ), 200 ppb, as indicated by the results of the hands-on training workshop. However, this study intended to compare the capability and performance of all participating laboratories. Therefore, all laboratories reported the actual amount of iodine in all test materials. The collaborative study of Jerse et al. about iodine determination in feed by ICP-MS also found S_L_ of approximately 35% when the iodine content in the test sample was less than 1 ppm [35]. In addition, a closed-group discussion with Lab 08 concerning corrective action indicated that the high *S_L_* in sample code MU-ETH I07 could have been due to the ICP-MS equipment having had problems during data determination.

To increase the efficacy and accuracy of an iodine monitoring program, a national or international proficiency testing (PT) program for iodine determination in direct-iodized sauces should be regularly organized at least once a year. Internal corrective actions and/or recruitment of consulting expertise should be undertaken for laboratories with unacceptable results in the PT program. Moreover, using the current national standard (2–3 ppm), iodine monitoring in the mandatory direct-iodized sauce is limited when the iodine test result is weighed by the adjusted maximum *S_L_* (±22%) based on this study. To meet the current standard, the iodine content in iodized products should be at least 2.6 ppm. However, the iodine content of a product will be 3.2 ppm when the test result has a positive bias (+22%). The prospective standard will range from 1.6 to 3.4 ppm when the Thai FDA’s judgment range is expanded by the adjusted maximum *S_L_* (±22%). The potential contribution of iodine intake among Thai non-pregnant adults from household salt, as per the nine processed foods that contribute the most to daily iodine intake and direct-iodized sauce, will be 106–126% of the recommended nutrient intake (RNI) and 27–32% of tolerable upper intake (UL) when calculated based on the assumption of a previous study [19] and prospective standard. In addition, this study’s maximum *S_L_* for iodine greater than method LOQ is 14%. Consequently, the Thai FDA’s judgment range for official control activities for iodine concentration in direct-iodized sauce should be expanded by 14–22% of the test result from each laboratory and have test results acceptable to the PT program and/or those of an accredited laboratory. The judgment range should also be revised and tightened following the standard set by yearly PT program results to be more precise and practical for implementation.

## 5. Conclusions

All target laboratories participated in a hands-on training workshop and two rounds of interlaboratory comparison for iodine analysis in direct-iodized sauce. The capacity of service laboratories increased via the hands-on training workshop in this study. Most of the participating laboratories (7/8) achieved a satisfactory performance (|z| < 2.0) for all six test samples from two rounds of interlaboratory comparison. Sample extraction using TMAH and ICP-MS detection with 6% 2-propanol and 0.01% triton X-100 as an internal standard can be used for iodine determination in direct-iodized sauce. After outlier removal, the reproducibility standard deviation (*S_L_*) varied from 7 to 22% at an iodine level of 0.03–4.81 ppm. Moreover, the Thai FDA’s judgment range for official control activities should be expanded by at least 22% (up to 41% based on reproducibility SD) to increase the efficacy and accuracy of an iodine monitoring program for the mandatory direct-iodized sauce.

## Figures and Tables

**Figure 1 foods-12-03513-f001:**
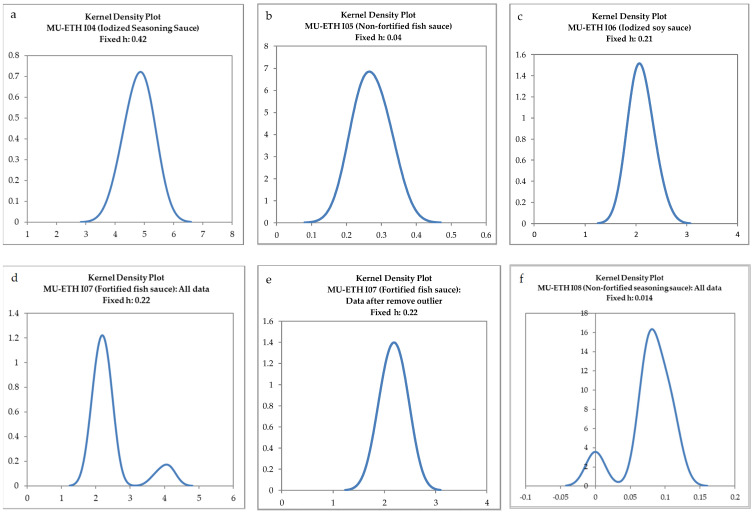
Kernel density plots of all test materials: (**a**) MU-ETH I04, (**b**) MU-ETH I05, (**c**) MU-ETH I06, (**d**) MU-ETH I07 (all data), (**e**) MU-ETH I07 (data removing outlier), (**f**) MU-ETH I08 (all data), (**g**) MU-ETH I08 (data removing outlier), (**h**) MU-ETH I09 (all data), and (**i**) MU-ETH I09 (data removing outlier).

**Figure 2 foods-12-03513-f002:**
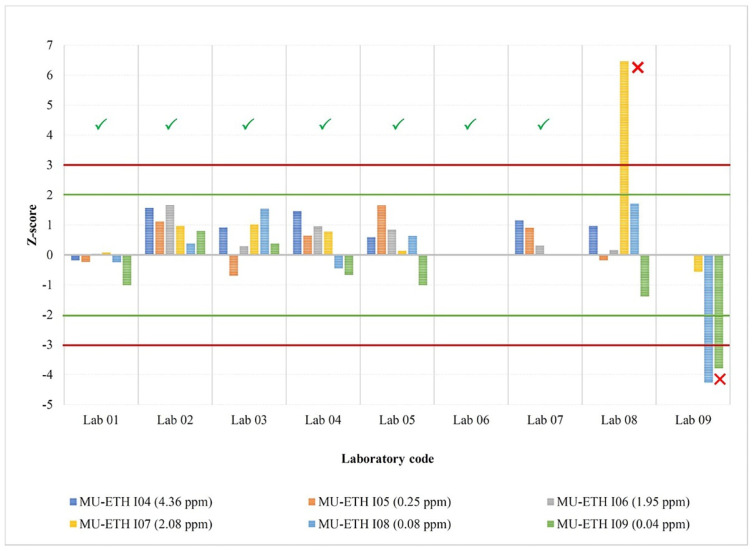
*Z*-score based on the reference lab (Lab 06) as ETH’s mean (*x*_pt_) and standard deviation (σ_pt_) from the Horwitz equation [28]: |*z*| ≤ 2, satisfactory (acceptable) result (✓); 2 ≤ |*z*| ≤ 3, questionable (warning) result; |*z*| > 3, unsatisfactory (unacceptable) result (☓). First-round study samples (ETH’s mean iodine content): MU-ETH I04 (4.36 ppm), MU-ETH I05 (0.25 ppm), and MU-ETH I06 (1.95 ppm). Second-round study samples (ETH’s mean iodine content): MU-ETH I07 (2.08 ppm), MU-ETH I08 (0.08 ppm), and MU-ETH I09 (0.04 ppm). Green line indicated satisfactory result (|*z*| ≤ 2) whereas red line indicated unsatisfactory results (|*z*| ≥ 3).

**Figure 3 foods-12-03513-f003:**
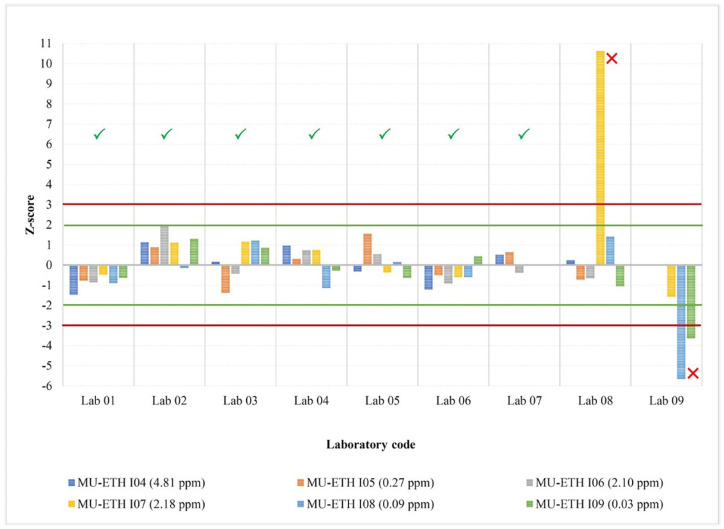
*Z*-score based on the consensus-assigned mean (*x_pt_*) and standard deviation (*σ_pt_*): |*z*| ≤ 2, satisfactory (acceptable) result (✓); 2 ≤ |*z*| ≤ 3, questionable (warning) result; |*z*| > 3, unsatisfactory (unacceptable) result (☓). First-round study samples (assigned mean iodine content): MU-ETH I04 (4.81 ppm), MU-ETH I05 (0.27 ppm), and MU-ETH I06 (2.10 ppm). Second-round study samples (assigned mean iodine content): MU-ETH I07 (2.18 ppm), MU-ETH I08 (0.09 ppm), and MU-ETH I09 (0.03 ppm). Green line indicated satisfactory result (|*z*| ≤ 2) whereas red line indicated unsatisfactory results (|*z*| ≥ 3).

**Table 1 foods-12-03513-t001:** Details of test materials.

Study Round	Sample Code	Description	Manufacture	Sample Collection Sites	Brand and Lot/MPG
1st	MU-ETH I04	Commercial iodized seasoning sauce added more potassium iodate solution during the sample preparation	Yan Wal Yun Co., Ltd., Samut Sakhon, Thailand	Supermarket	Healthy boy; 21102AFA1L 13:47
	MU-ETH I05	Non-iodized fish sauce	Rayong Fish Sauce Industry Co., Ltd., Rayong, Thailand	Factory	1st grade of tank no. 220
	MU-ETH I06	Iodized soy sauce	Yan Wal Yun Co., Ltd., Samut Sakhon, Thailand	Supermarket	Healthy boy; 21311ABA26 16:30
2nd	MU-ETH I07	Iodized fish sauce	Tang Sang Hah Co., Ltd., Chon Buri, Thailand	Supermarket	Tiparos; 10:03 (4025)
	MU-ETH I08	Iodized seasoning sauce	Thai Theparos PLC, Samut Prakan, Thailand	Supermarket	Golden Mountain; 16:12 (89)
	MU-ETH I09	Non-iodized soy sauce	Thai-Sino Food Co., Ltd., Samut Sakhon, Thailand	Factory	10/8/22

**Table 2 foods-12-03513-t002:** Homogeneity and stability of all test materials.

	MU-ETH I04	MU-ETH I05	MU-ETH I06	MU-ETH I07	MU-ETH I08	MU-ETH I09
Homogeneity mean	5.036	0.267	2.069	2.218	0.083	0.029
*s_s_*	0.142	0.026	0.100	0.106	0.009	0.004
σ*_pt_*	0.632	0.052	0.297	0.315	0.019	0.008
0.3σ*_pt_*	0.190	0.016	0.089	0.094	0.006	0.002
$s_{s}\leq 0.3\sigma_{pt}$	Adequately homogeneous	-	-	-	-	-
$\sqrt{c}$	-	0.027	0.129	0.155	0.011	0.004
$s_{s}\leq \sqrt{c}$	-	Sufficiently homogeneous	Sufficiently homogeneous	Sufficiently homogeneous	Sufficiently homogeneous	Sufficiently homogeneous
Stability mean	4.974	0.261	2.064	2.252	0.080	0.033
|$\bar{y}$ _1_ − $\bar{y}$_2_|	0.062	0.006	0.005	0.035	0.004	0.004
|$\bar{y}$_1_ − $\bar{y}$_2_| $\leq 0.3\sigma_{pt}$	Adequately stable	Adequately stable	Adequately stable	Adequately stable	Adequately stable	Adequately stable

**Table 3 foods-12-03513-t003:** List of participating laboratories.

Expert laboratory	Institute of Food, Nutrition and Health, Swiss Federal Institute of Technology Zurich (ETH), Switzerland
Government laboratories	Bureau of Nutrition, Ministry of Public Health (MOPH), Thailand
	Bureau of Quality and Safety of Food (BQSF), Department of Medical Sciences, Thailand
	Institute of Nutrition, Mahidol University (INMU), Thailand
	National Institute of Metrology of Thailand (NIMT), Thailand
Private laboratories	Central Laboratory (Thailand), Thailand
	ALS laboratory (group) Thailand, Thailand
	SGS Thailand, Thailand

**Table 4 foods-12-03513-t004:** Details of extraction method of participating laboratories.

Lab Code	Accredited ISO 17025	Sample Weight (g)	Chemicals Used for Extraction	Digestion Method	Condition (Temp. and Time)	Digestion (Round)
Lab 01	No	2.0	5% TMAH *	Oven	90 °C, 3 h	4 rounds
Lab 02	No	0.5–0.6	5% TMAH	Oven	90 °C, 3 h	1 round
Lab 03	No	0.5	5% TMAH	Oven	90 °C, 3 h	1 round
Lab 04	No	0.5	5% TMAH	Oven	90 °C, 3 h	1 round
Lab 05	Yes	0.5	4% TMAH	Oven	90 °C, 3 h	1 round
Lab 06	No	0.3	5% TMAH	Oven	85 °C, 3 h	1 round
Lab 07	No	0.5	5% TMAH	Oven	90 °C, 3 h	1 round
Lab 08	No	0.5	5% TMAH	Oven	90 °C, 3 h	1 round
Lab 09	No	0.5	5% TMAH	Oven	90 °C, 3 h	1 round

***** Tetramethylammonium hydroxide.

**Table 5 foods-12-03513-t005:** Details of ICP-MS conditions of participating laboratories.

Lab Code	Detail of the Instrument	Internal Standard Used
Lab 01	Triple Quadrupole ICP-MS (Agilent 8800, Agilent Technologies, Inc., Santa Clara, CA, USA)	2 ppb rhodium
Lab 02	Triple Quadrupole ICP-MS (PerkinElmer^®^’s NexION 2000, PerkinElmer, Inc., Waltham, MA, USA)	1st study round: 2 ppb rhodium 2nd Study round II: 10 ppb tellurium
Lab 03	Single Quadrupole ICP-MS (Agilent 7900, Agilent Technologies, Inc., Santa Clara, CA, USA)	2 ppb rhodium
Lab 04	Single Quadrupole ICP-MS (Agilent 7900, Agilent Technologies, Inc., Santa Clara, CA, USA)	10 ppb tellurium
Lab 05	Single Quadrupole ICP-MS (Agilent 7900, Agilent Technologies, Inc., Santa Clara, CA, USA)	10 ppb tellurium
Lab 06	Multicollector (MC) ICP-MS (Thermo Fisher Scientific, Thermo Fisher Scientific Inc., Waltham, USA), Single Quadrupole ICP-MS (Thermo Fisher Scientific iCap RQ, Thermo Fisher Scientific Inc., Waltham, MA, USA)	10 ppb tellurium
Lab 07	Triple Quadrupole ICP-MS (PerkinElmer^®^’s NexION 2000, PerkinElmer, Inc., Waltham, MA, USA)	2 ppb rhodium
Lab 08	Triple Quadrupole ICP-MS (PerkinElmer^®^’s NexION 2000, PerkinElmer, Inc., Waltham, MA, USA)	1st study round: 2 ppb rhodium 2nd study round II: 10 ppb rellurium

**Table 6 foods-12-03513-t006:** Results of the assigned values in this study.

Study Round	Sample Code	The Portion of the Laboratory that Passed the Test (The Code of the Lab That Did Not Pass the Test)				Assigned Value (Mean ± SD) of Iodine Content * (ppm)	$\mathrm{Uncertainty}\;\mathrm{of}\;\mathrm{Assigned}\;\mathrm{Value}\;(\mathrm{ppm})\;(u_{x_{pt}})$
		Grubb’s Test	Dixon’s Test	Kernel and Scatter Plot	Cochran’s Test		
1st	MU-ETH I04	8/8	8/8	8/8	8/8	4.81 ± 0.38	0.133
	MU-ETH I05	8/8	8/8	8/8	8/8	0.27 ± 0.04	0.014
	MU-ETH I06	8/8	8/8	8/8	8/8	2.10 ± 0.16	0.057
2nd	MU-ETH I07	8/8	8/8	7/8 (Lab 08)	7/8 (Lab 08)	2.18 ± 0.17	0.061
	MU-ETH I08	8/8	8/8	7/8 (Lab 09)	8/8	0.088 ± 0.016	0.005
	MU-ETH I09	8/8	8/8	7/8 (Lab 09)	7/8 lab (Lab 05)	0.032 ± 0.009	0.003

* Outliers were removed before calculating the mean and SD for the assigned values.

**Table 7 foods-12-03513-t007:** Summary of standard deviation from all laboratories in the collaborative study.

Sample Code	Mean (ppm)	Standard Deviation of Repeatability (*S_r_*)		Standard Deviation of Reproducibility (*S_L_*)	
		(ppm)	(%)	(ppm)	(%)
MU-ETH I04	4.81	0.11	2.2	0.35	7.3
MU-ETH I05	0.27	0.01	5.5	0.04	14.3
MU-ETH I06	2.10	0.05	2.5	0.16	7.7
MU-ETH I07	2.18	0.05 (0.09) *	2.3 (4.0) *	0.17 (0.66) *	7.9 (30.4) *
MU-ETH I08	0.088	0.005 (0.006) *	6.2 (6.6) *	0.013 (0.034) *	15.0 (38.8) *
MU-ETH I09	0.032	0.006 (0.006) *	17.1 (18.3) *	0.007 (0.013) *	21.7 (40.7) *

***** Value in parenthesis included all results of all laboratories without the removal of any outlier.

## Data Availability

Data are contained within the article.

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
