# Peer review of "Collaborative Study for Iodine Monitoring in Mandatory Direct-Iodized Sauce in Thailand"

_foods, 2023, doi:10.3390/foods12183513_

Round 1

Reviewer 1 Report

This paper is certainly interesting. In the article the content of iodine salts in commercial samples with and without external additions of iodine salts is analyzed in different laboratories. In my opinion the most critical part of the article is related to the description of the conditions of the extraction and preparation of the samples for the ICP-MS measurements. Particularly due to the importance of these procedures in the evaluation of the experimental results. Furthermore, the characteristics of the different instruments used should be better detailed, the range of which goes from multicollector, to triple quadrupole to single quadrupole etc.

It should also be considered that this study in which the statistical aspects appear fundamental, requires repeating the measurements well beyond the triplicate, which is acceptable for other studies.

There are other details in the text that should be treated in a more formal way, such as the section:

"2.1. Hands-on training workshops.

Zurich, who had expertise in iodine determination...."

In my opinion it is always necessary to refer to the laboratory and not generically to "Zurich...".

For all these reasons I suggest that the work be subject to "Major revision".

Minor editing of English language are  required

Author Response

REVIEWER 1

Comments and Suggestions from the Reviewer

Response from the Author

This paper is certainly interesting. In the article the content of iodine salts in commercial samples with and without external additions of iodine salts is analyzed in different laboratories.

Thank you for your valuable comments and suggestions on our manuscript. We appreciate the time and effort you have dedicated to reviewing our work. 

In my opinion the most critical part of the article is related to the description of the conditions of the extraction and preparation of the samples for the ICP-MS measurements. Particularly due to the importance of these procedures in the evaluation of the experimental results.

Detailed information about the preparation of the test samples for ICP-MS measurements has been included in the revised manuscript.

Furthermore, the characteristics of the different instruments used should be better detailed, the range of which goes from multicollector, to triple quadrupole to single quadrupole etc. It should also be considered that this study in which the statistical aspects appear fundamental, requires repeating the measurements well beyond the triplicate, which is acceptable for other studies.

Detailed information about the ICP-MS machines have been added in the revised manuscript. Please refer to Table 4 in the revised version of the manuscript.

There are other details in the text that should be treated in a more formal way, such as the section:

"2.1. Hands-on training workshops.

Zurich, who had expertise in iodine determination...."

In my opinion it is always necessary to refer to the laboratory and not generically to "Zurich...".

We apologize for any errors in the section 2.1. We have re-written the first sentence of the first and second paragraphs in this section.

For all these reasons I suggest that the work be subject to "Major revision".

Thank you for your valuable comments and suggestions on our manuscript. 

Reviewer 2 Report

If the manuscript is considered suitable for the journal I have some specific comments on the text

Paragraph 2.2.1: it is not clear how the homogeneity assessment was performed. If I understand correctly five samples were analysed. Based on both ISO Guide 35 and ISO 13528 at least 10 samples should be tested (in duplicate per ISO 13528). The number of samples can be reduced with an adequate justification (previous study, bibliographic data, etc.). Furthermore, some details should be provided on the repeatability of the analytical method used for the assessment of material’s homogeneity. Furthermore, no information on the stability studies have been presented (at least for the duration of the study). The assessment of both homogeneity and stability is a key point to assure that the material is adequate and its quality does not affect the performance of the participating laboratories. In conclusion, this paragraph should be improved

Paragraph 2.2: Kernel density plot is used to check the unimodality of the data distribution rather than to detect outliers. Specifically, the presence of minor modes is linked to the bandwidth used for the test, so the value of bandwidth should be specified in respect to the standard deviation for proficiency assessment.

Please, remove “robust” z score because the z score is not robust, but “robust” can be the indicator of central tendency and the indicator of data dispersion

Paragraph 3.2.2: As for Kernel density plot I suggest to include at least one representative plot (2nd round) or to state in the manuscript “the unimodality was proven using Kernel density plot at a bandwidth equal to X% of standard deviation for proficiency assessment” or something like this

The assigned values should be considered together with their uncertainties (uxpt) and it should be specified if the criterion on the uncertainty is fulfilled (par. 9.2 of ISO 13528). This should be done for the assigned values based both on reference laboratory and on consensus from participants

Paragraph 3.2.3: the procedure to extrapolate Sr and SR should be briefly described

Figure 1: Is Lab06 missing in the figure or are all z-scores equal to zero? In the latter it should be specified in the graph or in the figure caption

Paragraph 4: It is not clear the sentence “That amount…..curve”. Does it mean that laboratories using matrix-matched calibration have a LoQ greater than laboratories using external calibration? As for this specific aspect were laboratories free to choose the calibration approach?

Paragraph 2.1: The first sentence should be re-written since it seems incomplete (who “gave” what?).  In the sentence “During…….step” the verb is missing (gave?)

Author Response

REVIEWER 2

Comments and Suggestions from the Reviewer

Response from the Author

Paragraph 2.2.1: it is not clear how the homogeneity assessment was performed. If I understand correctly five samples were analysed. Based on both ISO Guide 35 and ISO 13528 at least 10 samples should be tested (in duplicate per ISO 13528). The number of samples can be reduced with an adequate justification (previous study, bibliographic data, etc.). Furthermore, some details should be provided on the repeatability of the analytical method used for the assessment of material’s homogeneity. Furthermore, no information on the stability studies have been presented (at least for the duration of the study). The assessment of both homogeneity and stability is a key point to assure that the material is adequate and its quality does not affect the performance of the participating laboratories. In conclusion, this paragraph should be improved.

Thank you for your valuable comments and suggestions on our manuscript. Additional information on homogeneity and stability testing has been added in section 2.2.2 Statistical analysis using the ISO 13528:2022 for statistical testing for all steps including homogeneity, stability, uncertainty of assigned value, assigned value, and evaluation.

Paragraph 2.2: Kernel density plot is used to check the unimodality of the data distribution rather than to detect outliers. Specifically, the presence of minor modes is linked to the bandwidth used for the test, so the value of bandwidth should be specified in respect to the standard deviation for proficiency assessment.

Please, remove “robust” z score because the z score is not robust, but “robust” can be the indicator of central tendency and the indicator of data dispersion.

- I agree with the suggestion, and the related sentences have been revised.  The bandwidth was set at 0.75spt according to ISO 13528:2022. The minor mode was shown apart from the unimodal curve.

- The term “robust” from the term “z score” was removed throughout the manuscript.

Paragraph 3.2.2: As for Kernel density plot I suggest to include at least one representative plot (2nd round) or to state in the manuscript “the unimodality was proven using Kernel density plot at a bandwidth equal to X% of standard deviation for proficiency assessment” or something like this.

The assigned values should be considered together with their uncertainties (uxpt) and it should be specified if the criterion on the uncertainty is fulfilled (par. 9.2 of ISO 13528). This should be done for the assigned values based both on reference laboratory and on consensus from participants.

- As per the reviewer’s suggestion, Kernel density plot of all test materials (including all data and data remove outlier) have been added in Figure 1. The optimum bandwidth is set at 0.75spt according to ISO 13528:2022.

- Uncertainties regarding assigned values for each test material have been included in Table 6 of revised manuscript.

Paragraph 3.2.3: the procedure to extrapolate Sr and SR should be briefly described.

Figure 1: Is Lab06 missing in the figure or are all z-scores equal to zero? In the latter it should be specified in the graph or in the figure caption.

- Lab 06 is the reference laboratory; therefore the z score is equal to 0 which the graph does not show. The legend for lab 06 was added.

Paragraph 4: It is not clear the sentence “That amount…..curve”. Does it mean that laboratories using matrix-matched calibration have a LoQ greater than laboratories using external calibration? As for this specific aspect were laboratories free to choose the calibration approach?

This sentence has been rewritten. That amount is below the method limit of quantification (LOQ), 200 ppb, as indicated in the results of the hands-on training workshop. However, this study intended to compare the capability and performance of all participating laboratories. Consequently, all laboratories reported the actual amounts of iodine in all test materials.

Paragraph 2.1: The first sentence should be re-written since it seems incomplete (who “gave” what?).  In the sentence “During…….step” the verb is missing (gave?)

We apologize for any errors in section 2.1. We have rewritten the first sentence of the first and second paragraphs in this section.

Reviewer 3 Report

The paper is well written and present and interesting collaborative study on the determination of iodine after alkaline extraction. The novelty of the study might be limited, as there is no new methodology or new results in relation to iodine intake, but the research and information presented is important seen with respect on how to improve the quality of the analytical chemistry applied for the analysis of food products, which is important for society and the well-being of the population. My recommendation is that the paper need some correction and editing, see below.

In the introduction it is argued that iodine determination in salty sauces are difficult and the better methodologies are needed, this is no doubt correct, but the argument that a difference in iodine content in fish sauce between 1,9 and 3,3 ppm is crucial and may led to negative impacts on costs etc is exaggerated. First of all the difference of 1,4 ppm is hardly crucial, secondly the information is anecdotal, and not part of a systematic survey or investigation. That section should be rewritten.

Many different statistics test are applied and referenced, but they should be better explained, and it possible maybe a few examples could be shown e.g. for the outlier test. As an example it is mentioned in the beginning of the results section that alkaline extraction passed the accuracy test, and the microwave digestion did not, but there is no explanation on how the accuracy test was carried out, and what statistics exactly was used.

Likewise, kernel density plot, grubbs test and many more are mentioned, but no data are shown, please show more data and examples on how the statistical test were applied.

In the discussion, it is mentioned that the problems with I08 and I09, which contain low levels of iodine, could be that the amount is below the LOQ of some of the labs. Why the “could be”, I assume that for all laboratories to participate, they have to demonstrate that LOD and LOQ are adequate, so please site the actual estimated LOD and LOQ. It could be more information to state the iodine concentration actually measured on the ICPMS for these samples (after the sample prep of alkaline extraction), and compare this value with the LOD and LOQ of the different labs.

The English language is fine, only minor corrections needed

Author Response

REVIEWER 3

Comments and Suggestions from the Reviewer

Response from the Author

The paper is well written and present and interesting collaborative study on the determination of iodine after alkaline extraction. The novelty of the study might be limited, as there is no new methodology or new results in relation to iodine intake, but the research and information presented is important seen with respect on how to improve the quality of the analytical chemistry applied for the analysis of food products, which is important for society and the well-being of the population. My recommendation is that the paper need some correction and editing, see below.

Thank you for your valuable comments and suggestions on our manuscript. We appreciate the time and effort you have dedicated to reviewing our work. 

In the introduction it is argued that iodine determination in salty sauces are difficult and the better methodologies are needed, this is no doubt correct, but the argument that a difference in iodine content in fish sauce between 1,9 and 3,3 ppm is crucial and may led to negative impacts on costs etc is exaggerated. First of all the difference of 1,4 ppm is hardly crucial, secondly the information is anecdotal, and not part of a systematic survey or investigation. That section should be rewritten.

We have rewritten this section. Please refer to the fifth paragraph of the introduction in the revised version of the manuscript.

Many different statistics test are applied and referenced, but they should be better explained, and it possible maybe a few examples could be shown e.g. for the outlier test. As an example it is mentioned in the beginning of the results section that alkaline extraction passed the accuracy test, and the microwave digestion did not, but there is no explanation on how the accuracy test was carried out, and what statistics exactly was used.

- Information on statistics used for each step (homogeneity, stability, uncertainty, assigned value, etc.) have been added in the revised manuscript.

- For accuracy, the t-test comparison mean with the known value was used as the criteria for accepting the accuracy of certified reference material as mentioned in section 3.2 Results of the Hands-on Training Workshop

Likewise, kernel density plot, grubbs test and many more are mentioned, but no data are shown, please show more data and examples on how the statistical test were applied.

Kernel density plot of all test materials (including all data and data remove outlier) have been added in Figure 1. The optimum bandwidth was set at 0.75spt according to ISO 13528:2022.  Additional statistical information has also been added, e.g., Grubb’s test, Dixon’s test.

In the discussion, it is mentioned that the problems with I08 and I09, which contain low levels of iodine, could be that the amount is below the LOQ of some of the labs. Why the “could be”, I assume that for all laboratories to participate, they have to demonstrate that LOD and LOQ are adequate, so please site the actual estimated LOD and LOQ. It could be more information to state the iodine concentration actually measured on the ICPMS for these samples (after the sample prep of alkaline extraction), and compare this value with the LOD and LOQ of the different labs.

This sentence has been rewritten. The amount is below the method limit of quantification (LOQ), 200 ppb as indicated in the results of the hands-on training workshop. However, this study intended to compare the capability and performance of all participating laboratories Therefore, all laboratories reported the actual amount of iodine in all test materials. LOD and LOQ of each laboratory are not included in this study. However, the laboratory will use LOD and LOQ in the future for reporting iodine content.

Reviewer 4 Report

Iodine determination in salty condiment sauces by inductively coupled plasma mass spectrometry – results of a collaborative study

My decision: Major Revised

The authors of this paper analyzed the determination of iodine content in condiment sauces by ICP-MS in 8 laboratories. The authors have done a lot of related data collection and analysis work. However, many problems still need to be solved. I think it needs to solve some questions below:

1.     The innovation and the importance of this work should be clearly highlighted in the abstract, introduction, and conclusions. Please work on this and prove to us why this work is valuable. Would you explicitly specify the novelty of your work? What progress against the most recent state-of-the-art similar studies was made?

2.     Title: Title should be more concise and attractive to the reader.

3.     Abstract: The innovation and the importance of this work should be clearly highlighted in the abstract. Why choose ICP-MS and what are the advantages of ICP-MS?

4.     Keywords: Usually there are no abbreviations in keywords.

5.     Introduction: The literature review needs more updating work to provide a clear and concise up-to-date analysis. A review of the advantages and disadvantages of ICP-MS and other methods for iodine determination. The introduction could be better organized. While the general introduction is acceptable, the latest review is difficult to understand and does not infer specific ideas. What can this study do for other countries with insufficient and excessive iodine intake?

6.     Material and methods: It is suggested to add information on ICP-MS.

7.     Discussion: There is no sufficient discussion about the advantages of ICP-MS.

8.     Conclusion: Please make sure your conclusions section underscores the scientific value-added of your paper and the applicability of your findings/results. Highlights the novelty of your study. Highlight the significance and prospect of the research.

Author Response

REVIEWER 4

Comments and Suggestions from the Reviewer

Response from the Author

My decision: Major Revised

The authors of this paper analyzed the determination of iodine content in condiment sauces by ICP-MS in 8 laboratories. The authors have done a lot of related data collection and analysis work. However, many problems still need to be solved. I think it needs to solve some questions below:

Thank you for your valuable comments and suggestions on our manuscript. We appreciate the time and effort you have dedicated to reviewing our work. 

1.   The innovation and the importance of this work should be clearly highlighted in the abstract, introduction, and conclusions. Please work on this and prove to us why this work is valuable. Would you explicitly specify the novelty of your work? What progress against the most recent state-of-the-art similar studies was made?

We have added additional information and rewritten the abstract, introduction, discussion, and conclusion.

2.   Title: Title should be more concise and attractive to the reader.

We have changed the title to “Collaborative study for iodine monitoring in mandatory direct-iodized sauce in Thailand”.

3.   Abstract: The innovation and the importance of this work should be clearly highlighted in the abstract. Why choose ICP-MS and what are the advantages of ICP-MS?

We have rewritten the abstract, introduction, and discussion including the advantage of ICP-MS.

4.   Keywords: Usually there are no abbreviations in keywords.

We have changed the abbreviation to the full form. Please refer to the keyword in the revised version of the manuscript.

5.   Introduction: The literature review needs more updating work to provide a clear and concise up-to-date analysis. A review of the advantages and disadvantages of ICP-MS and other methods for iodine determination. The introduction could be better organized. While the general introduction is acceptable, the latest review is difficult to understand and does not infer specific ideas. What can this study do for other countries with insufficient and excessive iodine intake?

- We have added additional information about the efficiency comparison of ICP-MS and other methods in the introduction. Please refer to the fifth paragraph in the revised version of the manuscript.

- This study can apply this analytical method for these types of samples, of which fish sauce and soy sauce have been used in Vietnam, China, and Asian countries.

6.   Material and methods: It is suggested to add information on ICP-MS.

We have added additional details about the ICP-MS machine. Please refer to Table 4 in the revised version of the manuscript.

7.   Discussion: There is no sufficient discussion about the advantages of ICP-MS.

The advantages of ICP-MS have been added in the discussion.

8.   Conclusion: Please make sure your conclusions section underscores the scientific value-added of your paper and the applicability of your findings/results. Highlights the novelty of your study. Highlight the significance and prospect of the research.

The conclusion section has been rewritten in the revised manuscript. The novel of this study is the repeatability and reproducibility of iodine analysis by ICP-MS in direct-iodized sauces.  The output from this study was used by the Thai FDA to adjust the suitable range.  The committee responsible for is discussing the next steps for further official mandate.

Round 2

Reviewer 2 Report

All my comments were accepted and the document was revised appropriately taking into consideration my suggestions

Reviewer 4 Report

I have no more queries about this revised article additionally the authors have lately been improved the overall quality of their MS which is looking good and I am satisfied with the revised version of the MS. Thus it can be accepted for publication by this esteemed journal as it is.